# RESpecBench: How reliable is LLM-as-a-judge? Rigorous Evaluation of Specification Generation with Automated Verification

## Abstract

Large Language Models (LLMs) are increasingly used to assist formalization of natural language statements into formal specifications. Unlike syntax correctness, validating semantic correctness is particularly challenging and LLM-as-a-Judge has become the dominant assessment methodology due to its ease of use and great flexibility. However, the reliability of LLM-as-a-Judge has rarely been systematically evaluated. We introduce RESpecBench, a multi-domain benchmark with a *sound* and *automated* verifier, measuring the LLM's ability to produce precise, semantically equivalent specifications from informal natural language descriptions. RESpecBench spans five different domains, including Grade-School Math (GSM-Symbolic+), SQL, First-Order Logic (FOL), regular expressions (RegEx), and Rocq Prover tasks. We evaluate several state-of-the-art LLMs on RESpecBench and compare our sound verifier to LLM-as-a-Judge pipelines, demonstrating that LLM-as-a-Judge produces unreliable verdicts and substantially overestimates specification correctness. RESpecBench enables rigorous, automated, and sound evaluation of natural language into formal specification translation across multiple domains, ensuring formalized statements target the intended natural language properties.

## 1 Introduction

Large Language Models (LLMs) are now extensively used for software development, having been integrated into every day tools for software developers including editors such as VS Code with GitHub Copilot, and with CLI assistants like Codex and Claude Code. This rise in popularity has inspired a wave of code-generation benchmarks, such as (Jain et al., 2024; Jimenez et al., 2024; Zhuo et al., 2025; Chen et al., 2021; Austin et al., 2021). Yet, as with human-written code, there exists no guarantee that the written code is correct, and developers often use testing methods such as unit testing, integration testing, fuzzing, or property-based testing to increase their confidence in correctness.

Software verification addresses this gap by guiding the programmer to prove that a program satisfies a formal specification via program verifiers such as Dafny (Leino, 2010), Why3 (Filliâtre & Paskevich, 2013), Verus (Lattuada et al., 2023), or interactive theorem provers (ITPs) such as Rocq Prover (Bertot & Castéran, 2004) (formerly Coq proof assistant) and Lean Theorem Prover (Moura & Ullrich, 2021). These tools take in *specifications* to operate, which describe the intended behaviour of the program, and the programmer proves that their code conforms to the specification.

While software verification tools have strong guarantees and produced countless of successful software such as CompCert (Leroy, 2009), HACL* (Zinzindohoué et al., 2017), and seL4 kernel (Klein et al., 2009), they are widely considered to be time consuming and hard to use, and are often substituted by testing. Moreover, software verification is only meaningful if the underlying *specification* precisely captures the intended behaviours in the natural language (NL) statement. Writing such specifications are difficult, but are as important as writing the correctness proof itself: NL is ambigious, and translating it into a precise, machine-checkable code is a major bottleneck. Incorrect specifications cannot be automatically caught by verification tools, and writing incorrect specifications risks missing or proving the incorrect properties.

To aid developers with software verification, recent autoformalization efforts (Yang et al., 2024; Endres et al., 2024; Liu et al., 2025; Misu et al., 2024; Sun et al., 2024; Ma et al., 2025; Cosler et al., 2023; Wu et al., 2024) have leveraged automated proof and specification generation, and several benchmarks (Deng et al., 2025; Loughridge et al., 2025; Thakur et al., 2025; Ye et al., 2025; Kamath et al., 2023) have accompanied these developments. However, existing evaluations either (i) primarily target proof generation, (ii) rely on evaluation methods that are not sound for checking specifications, such as property-based testing, or (iii) rely on the LLM to prove the accuracy of specifications or use LLM-as-a-Judge. These approaches miss counterexamples, or cannot formally prove that the LLM-generated specifications are correct, providing no semantic equivalence guarantees for accepted answers.

To address these shortcomings, we introduce `RESpecBench`, to our knowledge the first *multi-domain* benchmark focused on evaluating *natural language* to formal specification translation with a *sound* and *automated* verifier. `RESpecBench` measures whether an LLM can produce a specification that is semantically equivalent to the oracle specification across five domains: GSM-Symbolic+, SQL, First-Order Logic (FOL), Regular Expressions (RegEx), and Rocq Prover, domains which were chosen to evaluate semantically distinct specification generation tasks. For each of the domains, we introduce an automated verifier with a timeout limit, and return counterexamples where applicable.

We evaluate several state-of-the-art LLMs on `RESpecBench` and compare our sound verifier against LLM-as-a-Judge pipelines. We demonstrate that LLM-as-a-Judge systematically overestimates specification correctness, whereas our verifier provides semantically equivalent judgements.

In summary, our paper includes the following contributions:

1. **RESpecBench.** We present `RESpecBench`, a multi-domain benchmark targeting NL to formal specification translation tasks. `RESpecBench` covers five domains: GSM-Symbolic+, SQL, First-Order Logic (FOL), Regular Expressions (RegEx), and Rocq Prover tasks. Each task includes an oracle specification, an NL statement describing the problem, and additional properties required by the verifier.

2. **Efficient Sound Verifier.** For each domain in `RESpecBench`, we integrate an automated and sound verifier, which takes the LLM-generated specification and the oracle specification to output a verdict of `Success`, `Unknown`, or `Refuted`, with a counterexample model where applicable. The verifiers include: Z3 (De Moura & Bjørner, 2008) for GSM-Symbolic+ and FOL, DFA equivalence for RegEx, Polygon (Zhao et al., 2025) for SQL, and CoqHammer (Czajka & Kaliszyk, 2018) for Rocq Prover tasks. We provide details on the verifiers in Section 3, and provide assumptions in Section 2.2.

3. **Evaluation.** We evaluate several state-of-the-art LLMs on `RESpecBench` and compare our sound verifier with two different LLM-as-a-Judge setups. Across domains, we observe that LLM-as-a-Judge approach systematically overestimates the correctness of generated specifications.

## 2 BACKGROUND

### 2.1 VERIFICATION TOOLS

Interactive theorem provers such as Rocq Prover (Bertot & Castéran, 2004) and Lean 4 (Moura & Ullrich, 2021) require the programmer to encode the specification in their language, and write the proof of correctness manually, whereas Satisfiability Modulo Theories (SMT) solvers such as Z3 (De Moura & Bjørner, 2008) or CVC4 (Barrett et al., 2011)/CVC5 (Barbosa et al., 2022) operate on different theories and a boolean satisfiability solver to refute a formula across different domains such as arithmetic, arrays, bit vectors, etc. While being more powerful than SMT solvers, ITPs take major effort to use, and programmers often prefer automated tools powered by SMT solvers.

Software verification tools, or more generally known as automated theorem provers (ATPs), such as Dafny (Leino, 2010), Viper (Müller et al., 2016), Why3 (Filliâtre & Paskevich, 2013), VeriFast (Jacobs et al., 2011), Verus (Lattuada et al., 2023), KeY (Ahrendt et al., 2005), and VerCors (Blom & Huisman, 2014) automatically translate programs into SMT constraints, delegating proof obli-

gations to the underlying solvers. This abstracts the underlying complexity in proof verification, making them preferable over ITPs for software verification.

SMT solvers also power tools we use in `RESpecBench`. CoqHammer (Czajka & Kaliszyk, 2018) integrates into Rocq Prover, delegating proof obligations to multiple SMT solvers and first-order automated theorem provers such as Vampire (Bártek et al., 2025) and E Prover (Schulz, 2002), and reconstructs successful proofs in Rocq Prover. Polygon (Zhao et al., 2025) uses Z3 over a formally specified subset of SQL to verify query equivalence and disambiguation, providing a sound symbolic reasoning engine.

### 2.2 DEFINITIONS & ASSUMPTIONS

**Soundness.**  We define our verifier to be sound, meaning, for every oracle specification and the LLM-generated specification, if one of the verifiers outputs a verdict other than `Unknown`, then the verdict is guaranteed to indicate the semantic equivalence between the LLM-generated specification and the oracle specification.  We assume underlying SMT solvers as well as automated theorem provers used inside the verifiers to be sound, and that they will never return an incorrect verdict.

**Automation and efficiency.**  Our verifiers are *automated*, meaning they do not require human intervention or an LLM-aided proof to verify if the two specifications are semantically equivalent. Our verifiers are also *efficient*, and can terminate within a reasonable time limit (set as 4 seconds in our evaluations).

### 2.3 PREVIOUS WORK

**Code generation benchmarks.**  Recent developments in LLMs have made it necessary to evaluate how they perform in generating code.  Previous code generation benchmarks include MBPP Austin et al. (2021), HumanEval (Chen et al., 2021), LiveCodeBench (Jain et al., 2024), SWE-Bench (Jimenez et al., 2024), BigCodeBench (Zhuo et al., 2025), and many more.  These benchmarks rely on running tests hidden to the model to produce a judgement in evaluation, which has been shown to be not reliable by recent studies (Liu et al., 2023; Chowdhury et al., 2024).

**Software verification benchmarks.**  Previous work in software verification benchmarks primarily target verification tools, e.g., Dafny (Leino, 2010). These include CloverBench (Sun et al., 2024), MBPP-DFY (Misu et al., 2024), and DafnyBench (Loughridge et al., 2025). FVAPPS (Dougherty & Mehta, 2025), miniCodeProps (Lohn & Welleck, 2024), Verina (Ye et al., 2025), and Clever (Sun et al., 2024) target Lean 4 Theorem Prover (Moura & Ullrich, 2021), whereas VerifyThis-Bench (Deng et al., 2025) targets multiple verification languages.

Among prior work, only MBPP-DFY, CloverBench, VerifyThisBench, Clever, and Verina target specification generation. CloverBench checks semantic equivalence by asking Dafny to prove an equivalence between the generated and oracle specifications, which is a sound approach restricted in a single domain. VerifyThisBench relies on LLM-as-a-Judge to verify generated specifications. Clever asks the LLM to produce a Lean proof of equivalence, which is sound when the proof checks. However, the best model (GPT-4o with COPRA (Thakur et al., 2024)) succeeds on only 1.86% of the tasks. Verina evaluates correctness with test cases and property-based testing (PBT), which are not sound. Finally, MBPP-DFY and other studies such as (Fan et al., 2025) have evaluated specification generation by running the software verification tools, and manually inspecting the outputs to classify strengthened and weakened specifications.

In contrast, `RESpecBench` targets specification generation for multiple domains. For each one of the domains, `RESpecBench` provides a *sound* and *automated* verifier, none of which rely on the LLM to generate a proof of correctness. We demonstrate the summary of existing specification-generation benchmarks on Table 1.

**LLM-as-a-Judge evaluations.**  Prior work examines LLM-as-a-Judge from several angles: 1) preference against human queries (Zheng et al., 2023), 2) code knowledge and judgement (Zhao et al., 2024) 3) benchmarking across multiple domains (Tan et al., 2025).  Other studies identify systematic biases in LLM-as-a-Judge evaluations (Wang et al., 2023; Zheng et al., 2023), especially when benchmarking human versus LLM preferences.  Surveys also reinforce these findings (Gu

| Benchmark | LLM-independent? | Sound? | Automated? | Domain |
|---|---|---|---|---|
| MBPP-DFY | ✓ | ✓ | ✗ | Dafny |
| VerifyThisBench | ✗ | ✗ | ✓ | ATPs |
| CloverBench | ✓ | ✓ | ✓ | Dafny |
| Verina | ✓ | ✗ | ✓ | Lean |
| Clever | ✗ | ✓ | ✓ | Lean |
| RESpecBench | ✓ | ✓ | ✓ | Multi-domain |

Table 1: Trade-offs among existing benchmarks.

Table 2: `RESpecBench` data sources by domain, totalling 707 questions across 5 domains.

| Domain | Primary source(s) | Notes on source | Count |
|---|---|---|---|
| GSM-Symbolic+ | GSM-Symbolic (Mirzadeh et al., 2024), from GSM8K (Cobbe et al., 2021) | Template variants and perturbations on templates, with relevant constraints exposed to the LLM. | 250 |
| SQL | Polygon LeetCode Bench (Zhao et al., 2025); Scythe (Wang et al., 2017), forums. | Original NL statements taken from SQLTeam and LeetCode. | 76 |
| RegEx | NL-RX-Turk (DeepRegex) (Locascio et al., 2016) | NL refined to be more specific. | 200 |
| FOL | FOLIO (Han et al., 2024) | Subset taken by measured complexity (more constant count). | 150 |
| Rocq Prover | Subset of Clever (translated to Rocq Prover from Lean 4) | CoqHammer-friendly subset taken only. | 31 |

et al., 2025). However, most existing work compares judges across multiple domains, while overlooking the LLM's ability to judge their own outputs. To our knowledge, none of the studies directly examine how well LLMs judge specifications, which we largely believe is due to the lack of verifiable ground-truth specifications in previous benchmarks.

## 3 RESpecBench BENCHMARK

**Benchmarking pipeline.** Specification generation covers a wide range of formal languages and semantics, which creates two practical obstacles for automated evaluation. First, differences in syntax and semantics mean that each domain needs a specialized verifier. Second, many complex specification equivalence problems are undecidable in general, so automated checks may not terminate or may return inconclusive results within a reasonable time budget.

To address these constraints while preserving soundness and automation, `RESpecBench` targets domains that are either decidable or prover-friendly in practice, and pairs each domain with a specialized verifier, summarized in Figure 1. Additionally, we expose a timeout setting for each verifier, which controls the time taken before `Unknown` verdicts.

**Dataset curation, processing and verifiers.** We curate `RESpecBench` from publicly available sources and benchmarks, summarized in Table 2. Then, we integrate a verifier for each domain. In this section, we describe the additional processing applied to the data and verifiers.

**GSM-Symbolic+.** While GSM-Symbolic (Mirzadeh et al., 2024) provides templated variants of the original problems with varying numbers, they prompt the LLM with automatically generated new questions from the templates. From GSM-Symbolic templates, we extract variable names and associated type information (e.g., units, integer vs. real). For each template we construct a symbolic version of the problem with variables and expose relevant public constraints to the LLM, demonstrated with an example in Figure 2. We compile the response language with variables and

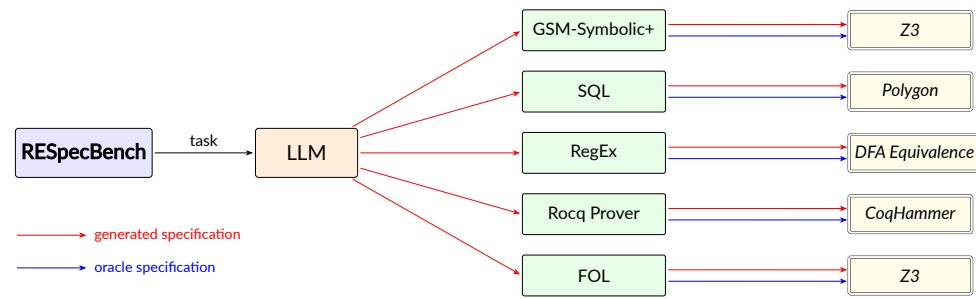

Figure 1: Overview of `RESpecBench` pipeline. Each natural language task is given to an LLM, which produces a candidate specification. The generated specification is compared against the oracle specification using the domain-specific verifiers.

conditionals to Z3 formulas, and Z3 either proves equivalence between the specifications, produces a counterexample model, or times out.

**SQL.** We pair each query with a table schema and the original natural language prompt taken from forums. As our verifier, we use Polygon, which translates SQL queries into Z3 formulas and checks equivalence by either proving them identical or producing a counterexample database.

**RegEx.** The NL-RegEx pairs in DeepRegex (Locascio et al., 2016) often contain ambiguous natural-language descriptions. To reduce ambiguity, we use a secondary language model, GPT-5 mini (OpenAI, 2025a) that rewrites the original NL prompt to be more specific, and manually verify the rewritten prompts. As for our verifier, we apply the DFA equivalence checker provided by DeepRegex. This approach is sound and complete for regular expressions, thus the verifier returns only `Success` or `Refuted` but never `Unknown`.

**Rocq Prover.** We select Rocq Prover problems that are compatible with CoqHammer's workflow (for example, avoiding recursive definitions that make it harder for ATPs to find a proof). From the dataset in Clever (Thakur et al., 2025), we manually select questions that are suitable for CoqHammer to Gallina (the language of Rocq Prover), and formulate an equivalence theorem between the original and the generated specification for CoqHammer to prove. In CoqHammer, the timeout is controlled by the `ATPLimit` parameter. However, in some cases, external ATPs succeed while proof reconstruction fails, in which case we still treat the result as `Success`, since the underlying ATPs are assumed to be sound. CoqHammer does not produce counterexamples, and the incorrect specifications result in `Unknown` rather than `Refuted`.

**FOL.** From each group of candidate sentences in FOLIO, we extract the NL sentence and the expert FOL formalization, discard examples that cannot be parsed reliably, and rank candidates by formula complexity (e.g., number of distinct constants). We then select the top 150 examples and blacklist sentences that are ambiguous or otherwise unsuitable. We define a grammar for the FOL language used in FOLIO and compile formulas to Z3. The verifier can either return a counterexample, establish equivalence, or time out, returning `Unknown`.

## 3.1 EVALUATION

We evaluate various state-of-the-art LLMs on `RESpecBench`, setting the temperature to 1.0 for each model, and using 'medium' reasoning effort for all applicable reasoning models. We report overall success rate as the *average* across the five domains. After prompting the models, we evaluate the responses by processing the output and passing them onto the verifiers, all of which are configured to have a 4-second timeout. We include additional details about each one of the models as well as our evaluation environment in Appendix A.

Our three evaluation categories are:

**GSM-Symbolic+**

**Natural Language:** {name} is feeding his livestock {food}. Each {animal1} needs {n1} {unit}, and each {animal2} needs {n2} {unit} less than twice the amount each {animal1} needs. If there are {k1} {animal1}s and {k2} {animal2}s, how many {unit} of {food} does {name} need?

**LLM-Exposed Constraints:** unit in ['kg', 'kilograms', 'pounds']

**Hidden Constraints:** $2 \cdot n1 - n2 > 0$

- - - - - - - - - - - - - - - - - - - - - - - - - - - - - - - - -

**Claude Sonnet 4:**
I need to solve this step by step.

Let me identify what each animal needs:
– Each {animal1} needs {n1} {unit}
– Each {animal2} needs {n2} {unit} less than twice the amount each {animal1} needs

```
lines omitted...
```

The total amount needed is the sum of all these amounts.

$(k1 \cdot n1) + (k2 \cdot ((2 \cdot n1) - n2)) + (k3 \cdot (n1 + ((2 \cdot n1) - n2)))$

**FOL**

**Natural Language:** All languages within a language family are related to each other.

**Symbols:** Related (2), MemberOf (2)

- - - - - - - - - - - - - - - - - - - - - - - - - - - - - - - - -

**Claude Sonnet 4:**
I need to formalize "All languages within a language family are related to each other."

Let me break this down:
– If two languages are both members of the same language family, then they are related to each other
– This means: for any two languages x and y, if there exists a language family z such that both x and y are members of z, then x and y are related

```
lines omitted...
```

$\forall x, y, z \, (MemberOf(x, z) \land MemberOf(y, z) \rightarrow Related(x, y))$

Figure 2: Two example questions and LLM responses, from GSM-Symbolic+ and FOL, respectively. In FOL symbols, the numbers in parentheses indicate the arity of the predicate.

- **Sound verifier.** Verifiers measure end-to-end performance for specification generation. The models are asked to output specifications in a formatted manner, which is then inputted to the verifiers. This evaluation forms the baseline for `RESpecBench`.

- **Formalization Judge.** Given the natural-language problem and a candidate specification from its previous responses, the LLM outputs a score between 0 to 3 (inclusive). We classify the output as correct when the score is greater than or equal to 2. Further details on scoring criteria and prompts are included in Appendix B.

- **Equivalence Judge.** Given the oracle and LLM-generated specifications and the constraints, the judge assigns a score of equivalence, as done with the Formalization Judge. To mitigate bias in the ordering of specifications in the prompt (Wang et al., 2023; Zheng et al., 2023), we employ an approach similar to that followed by JudgeBench (Tan et al., 2025), where the judge is prompted twice with a different order of specifications. If both prompted judges agree on the same verdict, we use that decision. Otherwise, we sum the two scores and classify the verdict as `Success` if the total of the scores is greater than or equal to 4.

Further, for both judges we classify a case as `Unknown` if the model outputs an invalid or malformed response (e.g., missing or out-of-range score). Such cases are rare and shown in detail in Appendix C, but we exclude these judgements from all metrics we use.

To compare judge decisions against the verifiers and evaluate the viability of LLM-as-a-Judge, we base our analysis on Inflation and the false acceptance rate (FAR) and false rejection rate (FRR) metrics, defined as follows (where # indicates the count):

$$\text{FAR} = \frac{\#\{\text{verifier} = \texttt{Refuted} \land \text{judge} = \texttt{Success}\}}{\#\{\text{verifier} = \texttt{Refuted} \land \text{judge} \neq \texttt{Unknown}\}}$$

$$\text{FRR} = \frac{\#\{\text{verifier} = \texttt{Success} \land \text{judge} = \texttt{Refuted}\}}{\#\{\text{verifier} = \texttt{Success} \land \text{judge} \neq \texttt{Unknown}\}}$$

$$\text{Inflation} = \text{Judge success rate (decided responses)} - \text{Verifier success rate (decided responses)}$$

## 3.2 RESULTS AND ANALYSIS

**Natural language understanding is the bottleneck.** Table 3 reports overall and per-domain verifier pass rates. Across models, the overall accuracy clusters near 50–55% while having higher scores on GSM-Symbolic+ and lower scores on Rocq and SQL.

Table 3: Verifier success rates over total tasks across domains and models.

| Model | GSM-Symbolic+ | SQL | FOL | RegEx | Rocq | Overall |
|---|---|---|---|---|---|---|
| GPT 5 | 88.4% | 32.9% | 67.3% | 66.5% | 16.1% | 54.3% |
| GPT 5 Mini | 85.2% | 31.6% | 67.3% | 70.5% | 16.1% | 54.1% |
| GPT-OSS 120B | 82.4% | 28.9% | 69.3% | 70.5% | 9.7% | 52.2% |
| GPT-OSS 20B | 82.0% | 28.9% | 64.0% | 67.5% | 9.7% | 50.4% |
| Claude Sonnet 4 | 70.0% | 34.2% | 68.7% | 69.0% | 9.7% | 50.3% |
| Gemini 2.5 Pro | 84.4% | 39.5% | 70.7% | 64.5% | 9.7% | 53.7% |
| Gemini 2.5 Flash | 76.0% | 35.5% | 68.0% | 53.0% | 12.9% | 49.1% |
| Qwen3-Next Thinking | 84.8% | 35.5% | 62.0% | 65.5% | 12.9% | 52.1% |
| Qwen3-Next | 80.4% | 32.9% | 60.0% | 62.0% | 6.5% | 48.3% |

Table 4: Overall judge performance for each model compared to overall verifier results.

| Model | Formalization Judge | | | Equivalence Judge | | |
|---|---|---|---|---|---|---|
| | FAR | FRR | Inflation | FAR | FRR | Inflation |
| GPT 5 | 84.2% | 11.5% | 18.7% | 14.1% | 23.0% | -11.6% |
| GPT 5 Mini | 86.2% | 5.5% | 23.1% | 20.8% | 5.9% | 1.9% |
| GPT-OSS 120B | 72.4% | 11.7% | 16.6% | 6.5% | 15.7% | -9.1% |
| GPT-OSS 20B | 69.2% | 9.0% | 16.4% | 10.1% | 19.0% | -9.7% |
| Claude Sonnet 4 | 76.9% | 7.6% | 22.5% | 16.0% | 16.1% | -3.6% |
| Gemini 2.5 Pro | 88.9% | 8.8% | 20.8% | 13.2% | 20.7% | -10.1% |
| Gemini 2.5 Flash | 91.1% | 4.0% | 31.5% | 26.8% | 15.0% | 2.0% |
| Qwen3-Next Thinking | 76.5% | 5.2% | 21.9% | 10.0% | 30.7% | -16.4% |
| Qwen3-Next | 82.3% | 8.6% | 25.2% | 14.4% | 23.9% | -9.8% |
| Mean | 80.9% | 8.0% | 21.9% | 14.6% | 18.9% | -7.4% |
| Median | 82.3% | 8.6% | 21.9% | 14.1% | 19.0% | -9.7% |

Table 4 further clarifies this gap with the overall results. The Formalization Judge adds, on average, $+21.9\%$ inflation to accuracy, indicating that models produce specifications that *look* compatible with the prompt but are often not semantically correct. In contrast, the Equivalence Judge yields *negative* overall inflation with a mean of $-7.4\%$, reflecting that, when asked to compare two specifications, judges are more conservative and frequently reject the generated specification.

Together, these patterns imply that the dominant failure mode lies in translating natural language requirements to neither relaxed, nor strict, constraints. Models can often recognize equivalence between formal specifications that they have formalized, but they frequently miss or misinterpret details in the natural-language statement, leading to a lower formalization success rate.

**Judges over-accept *'Is this specification correct?'* for their own answers.** Figure 3 shows that, when judging from natural language via Formalization Judge, models frequently accept incorrect or partial specifications, since FAR is high across domains. We believe that this reflects a plausibility bias in specifications understanding: the judge matches the semantics of the specification to the prompt, rather than verifying the specification's correctness. As an example, in our analysis of failures in SQL, which has the highest overall FAR, we find that typical misses include constraint details such as `NULL` field handling, which the Formalization Judge fails to account for.

By contrast, the Equivalence Judge is a more pessimistic estimator. Figure 4 shows negative or near-zero inflation on most domains, indicating systematic under-acceptance (higher FRR, as seen in Figure 3) when comparing two specifications. An exception is SQL, where Equivalence Judge still shows positive inflation for certain models.

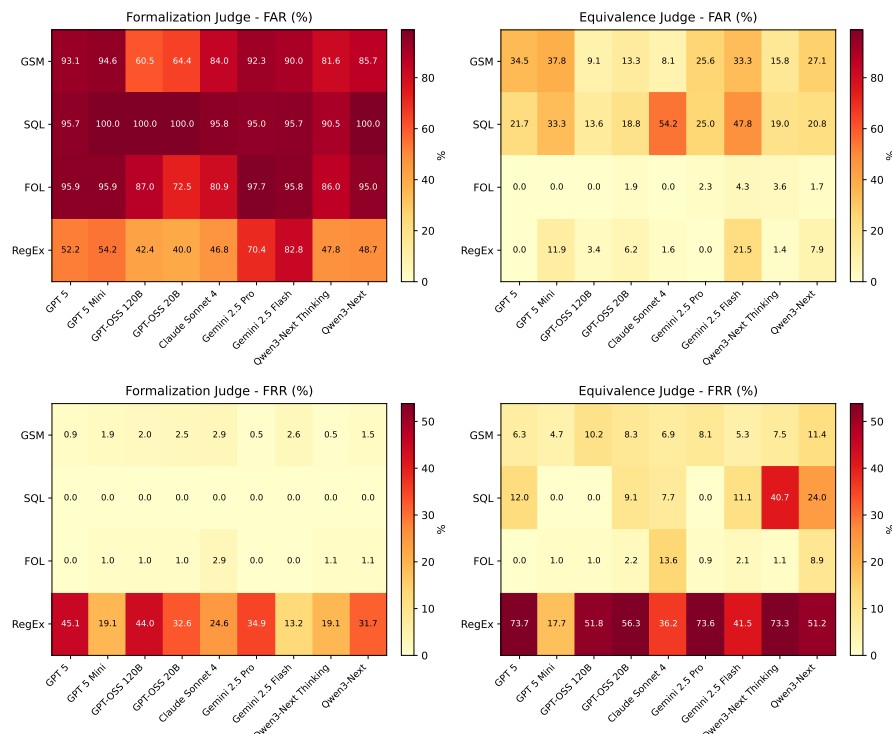

Figure 3: FAR and FRR rates for each model. Rocq Prover tasks are excluded as the domain does not yield `Refuted` results.

Formalization Judge is unsafe as an evaluator as it over-accepts plausible but wrong specifications. Equivalence Judge judging is more conservative but still not a perfect estimate for specification equivalence. For reliable evaluation, running verifiers remains a more accurate metric.

**Unreliable acceptances, reliable rejections.** Shown in Figure 3, the Formalization Judge has high FAR rates, frequently accepting incorrect, or partial specifications, indicating positive decisions are not necessarily trustworthy. By contrast, FRR is generally low, meaning that when the model's answers are rejected, the rejection is likely to be grounded. However, one notable exception is RegEx, where FRR spikes, and many semantically correct variants are rejected. We attribute this to the format sensitivity and ambiguity, where unlike SQL and FOL, our RegEx domain uses a custom language. These patterns show that equivalence evaluation reliability is also tied to the domain's semantics and the NL descriptions.

### 3.3 JUDGE AND VERIFIER SENSITIVITY

**Formalization Judge on oracle specifications.** Table 5 reports the Formalization Judge's overall success rate when scoring the oracle specifications against the natural language prompts. This rate is consistently higher than the end-to-end generation success rates, with the largest gains in SQL, FOL, and RegEx. This indicates that once the correct formalization is provided, the judge often recognizes it as plausible for certain domains.

At the same time, the judge is not a reliable validator on model outputs. Together with the low FRR with high FAR, this demonstrates an expected-answer bias: the judge tends to accept candidate specifications that match a phrasing it 'expects', and reject when the reference deviates from that phrasing, reagrdless of semantic correctness. Thus, NL-based judging still remains a weak filter, and not an option for judging specification correctness.

**Is automated verification timeout-sensitive?** In this experiment, we vary the solver timeout budget across durations ranging from 4 to 256s to test whether `Unknown` counts decrease with more

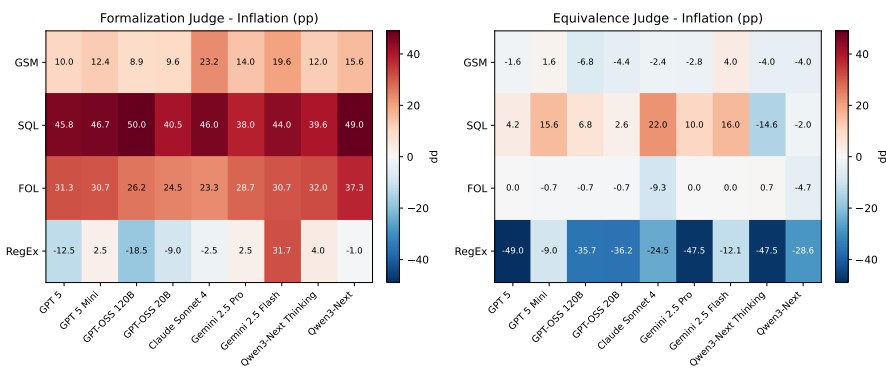

Figure 4: Overall inflation rates for each model and domain by percentage points (pp).

Table 5: Formalization Judge acceptance rates given oracle specifications across model and domains.

| Model | GSM-Symbolic+ | SQL | FOL | RegEx | Coq | Overall |
|---|---|---|---|---|---|---|
| GPT 5 | 63.6% | 72.4% | 83.3% | 17.5% | 67.7% | 60.9% |
| GPT 5 Mini | 69.2% | 80.3% | 79.3% | 55.5% | 61.3% | 69.1% |
| GPT-OSS 120B | 64.0% | 77.6% | 78.0% | 16.5% | 61.3% | 59.5% |
| GPT-OSS 20B | 64.0% | 75.0% | 77.3% | 36.5% | 61.3% | 62.8% |
| Claude Sonnet 4 | 66.4% | 75.0% | 80.0% | 33.5% | 74.2% | 65.8% |
| Gemini 2.5 Pro | 70.0% | 75.0% | 84.0% | 18.0% | 74.2% | 64.2% |
| Gemini 2.5 Flash | 74.4% | 81.6% | 88.7% | 45.0% | 77.4% | 73.4% |
| Qwen3-Next Thinking | 63.2% | 71.1% | 83.3% | 34.0% | 71.0% | 64.5% |
| Qwen3-Next | 64.4% | 77.6% | 79.3% | 34.5% | 64.5% | 64.1% |

time. For RegEx, GSM-Symbolic+, and FOL there are no timeouts under our budgets, so we focus on SQL and Rocq Prover using the best-performing models in each domain (Gemini 2.5 Pro for SQL, GPT-5 for Rocq Prover). Increasing the timeout from 4 to 15s only yields one fewer unknown for SQL, and 64s yields only one fewer for Rocq Prover. The remaining computations until 256s do not affect the unknowns. This suggests that increasing time budgets do not majorly affect `Unknown` results, and that timeouts are due to other limitations in ATPs. In short, automated checking is fast and feasible for RegEx, GSM-Symbolic+, and FOL, and exhibits strong diminishing returns with longer budgets on SQL and Rocq Prover, but still remains reliable for majority of the tasks.

## 4 CONCLUSION AND LIMITATIONS

We introduced `RESpecBench`, a multi-domain benchmark for natural language to specification generation with *efficient*, *sound* and *automated* checking across GSM-Symbolic+, SQL, FOL, RegEx, and Rocq Prover specifications. We evaluated several state-of-the-art LLMs on our benchmark and analyzed LLM-as-a-Judge reliability. Our results show that overall verified accuracy clusters in a narrow band across models, and that LLM-as-a-Judge is fragile for specification evaluation.

However, our work has several limitations. Some items remain `Unknown` on SQL tasks and Rocq Prover tasks do not return refutations, which are due to the limitations with the ATPs. Further, our Rocq Prover domain is limited with only 31 tasks, since it is difficult finding hard yet CoqHammer verifiable specifications. Additionally, we note that specification generation can be a vague task, given that natural language is highly open to interpretation.

We see three promising directions: (1) expanding decidable fragments and domains such as RegEx, (2) improving counterexample generation and ATPs for ITP specification verification, and (3) developing better models for rigorous specification generation. We hope `RESpecBench` serves as a reliable baseline and a driver for robust, verifiable specification synthesis.

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

# A   EVALUATION ENVIRONMENT

**Hardware.**   All verifier runs were executed on a 2018 MacBook Pro (Intel Core i5-8259U, 8 cores/8 threads, 2.3 GHz base), 8 GB RAM, and 16 GB swap.

**Operating system & Python.**   Ubuntu 24.04.3 (6.16.0-1-t2-noble). Python 3.12.3 in a virtual environment. We pin all Python dependencies in the artifact (`requirements.txt`).

**Docker image.**   To ensure reproducibility, our artifact includes a Docker image with all required solvers and theorem provers preinstalled, along with pinned versions and default settings. Because some ATPs are not distributed for ARM architectures, the image can only be ran on `x86_64` only.

Table 6: **Tool versions.**

| Component | Version / Build |
|---|---|
| Z3 | 4.11.2 (64-bit) |
| CVC4 | 1.8 (GCC 13.2) |
| E Prover | 3.0.03 (`c4e0c24`) |
| Vampire | 5.0.0 (`339063e1c`) |
| Coq | 8.20.0 |
| coq-stdlib | 8.20.0 |
| CoqHammer | 1.3.2+8.20 |
| Polygon (SQL) | as in (Zhao et al., 2025) |
| DFA equivalence (RegEx) | DeepRegex script |
| Python | 3.12.3 |

## A.1   LLM VERSIONS AND PARAMETERS

All generations and judge calls use temperature = 1.0, and other parameters as the provider default, and no tool/function calling. Our evaluations use OpenRouter API, and we provide the exact model strings below:

Table 7: Models used in the evaluations, including their model ID on OpenRouter.

| Model | API / Model ID |
|---|---|
| GPT-5 (OpenAI, 2025b) | `openai/gpt-5` (gpt-5-2025-08-07) |
| GPT-5 Mini (OpenAI, 2025a) | `openai/gpt-5-mini` (gpt-5-2025-08-07) |
| GPT-OSS 120B (OpenAI, 2025d) | `openai/gpt-oss-120b` |
| GPT-OSS 20B (OpenAI, 2025c) | `openai/gpt-oss-20b` |
| Claude Sonnet 4 (Anthropic, 2025) | `anthropic/claude-sonnet-4` (claude-sonnet-4-20250514) |
| Gemini 2.5 Pro (Google, 2025) | `google/gemini-2.5-pro` |
| Gemini 2.5 Flash (Google, 2025) | `google/gemini-2.5-flash` |
| Qwen3-Next 80B (Thinking, A3B) (Qwen Team, 2025c;b) | `qwen/qwen3-next-80b-a3b-thinking` |
| Qwen3-Next 80B (Instruct, A3B) (Qwen Team, 2025c;a) | `qwen/qwen3-next-80b-a3b-instruct` |

## B System Prompts for Scoring Criteria

This appendix provides the exact system prompts used for Formalization Judge and Equivalence Judge. The complete set of system prompts for all 15 domain-judge and domain-model combinations can be found in the artifact at `src/llm/system_prompts.py`.

For Formalization Judge, the model was given the following criteria:

```
- 0 indicates the answer is completely wrong,
- 1 indicates the answer is partially correct but has significant
errors, and won't pass the grading,
- 2 indicates the answer is mostly correct and should pass the
grading, but could be improved significantly,
- 3 indicates the answer is perfectly correct and matches the
expected result exactly.
```

The following examples were also provided to the judge:

```
Example valid responses:
```
0
```
Explanation:  The answer is completely wrong because it does not
address the problem at all, or the answer has a syntactical or
semantic error that makes it invalid with respect to the
requirements.
```
1
```
Explanation:  The answer has some correct elements but contains
significant errors that prevent it from being considered correct.
```
2
```
Explanation:  The answer is mostly correct and would likely pass
grading, but there are areas that could be improved significantly.

...  For this reason, I will give it a score of 3:
```
3
```
Explanation:  The answer is perfectly correct and matches the
expected result exactly.  The explanations are optional and were
included before the final answer.

Example invalid responses:
```
Score:  3
```
Explanation:  The answer must contain only the score as an integer
between 0 and 3, without any extra text.
```

For Equivalence Judge, the following criteria was given for scoring:

```
- 0 indicates the two responses are semantically completely
different or one of them have a syntax error,
- 1 indicates the two responses are semantically partially
equivalent but have significant errors, and won't pass the
grading,
- 2 indicates the two responses are semantically equivalent, but
could be improved significantly or have minor ambiguities,
```

– 3 indicates the two responses are perfectly semantically equivalent.

The following examples were also provided to the judge:

```
Example valid responses:
```
0
```
Explanation:  The two responses are completely different and not
semantically equivalent.
```
1
```
Explanation:  The two responses are semantically partially
equivalent but have significant errors, and won't pass the
grading.
```
2
```
Explanation:  The two responses are semantically equivalent, but
could be improved significantly or have minor ambiguities.
...  For this reason, I will give it a score of 3:
```
3
```
Explanation:  The two responses are perfectly semantically
equivalent.  The explanations are optional and were included
before the final answer.
Example invalid responses:
```
Score:  3
```
Explanation:  The answer must contain only the score as an integer
between 0 and 3, without any extra text.

## C    TOTAL UNKNOWN STATISTICS

This section shows the total `Unknown` results returned by any of the judges per domain. An `Unknown` result indicates a malformed response from the judge that could not be processed, and these cases are excluded from all reported metrics. While such occurrences are rare, we include these statistics for complete transparency and reproducibility.

Table 8: Total percentage of unknown cases by domain and model for each verifier.

| Model | GSM | SQL | FOL | RegEx | Coq |
|---|---|---|---|---|---|
| GPT 5 | 0.0% | 36.8% | 0.0% | 0.0% | 83.9% |
| GPT 5 Mini | 0.0% | 40.8% | 0.0% | 0.0% | 83.9% |
| Gemini 2.5 Pro | 0.0% | 34.2% | 0.0% | 0.0% | 90.3% |
| GPT-OSS 120B | 0.0% | 42.1% | 0.0% | 0.0% | 90.3% |
| Qwen3-Next Thinking | 0.0% | 36.8% | 0.0% | 0.0% | 87.1% |
| GPT-OSS 20B | 0.0% | 50.0% | 0.0% | 0.0% | 90.3% |
| Claude Sonnet 4 | 0.0% | 34.2% | 0.0% | 0.0% | 90.3% |
| Gemini 2.5 Flash | 0.0% | 34.2% | 0.0% | 0.0% | 87.1% |
| Qwen3-Next | 0.0% | 35.5% | 0.0% | 0.0% | 93.5% |

Table 9: Total percentage of unknown cases by domain and model for Formalization Judge.

| Model | GSM | SQL | FOL | RegEx | Coq |
|---|---|---|---|---|---|
| GPT 5 | 0.0% | 0.0% | 0.0% | 0.0% | 0.0% |
| GPT 5 Mini | 0.0% | 0.0% | 0.0% | 0.0% | 0.0% |
| Gemini 2.5 Pro | 0.0% | 0.0% | 0.0% | 0.0% | 0.0% |
| GPT-OSS 120B | 0.8% | 0.0% | 0.7% | 0.0% | 0.0% |
| Qwen3-Next Thinking | 0.0% | 0.0% | 0.0% | 0.0% | 0.0% |
| GPT-OSS 20B | 0.4% | 2.6% | 2.0% | 0.0% | 0.0% |
| Claude Sonnet 4 | 0.0% | 0.0% | 0.0% | 0.0% | 0.0% |
| Gemini 2.5 Flash | 0.0% | 0.0% | 0.0% | 0.5% | 0.0% |
| Qwen3-Next | 0.0% | 0.0% | 0.0% | 0.5% | 0.0% |

Table 10: Total percentage of unknown cases by domain and model for Equivalence Judge.

| Model | GSM | SQL | FOL | RegEx | Coq |
|---|---|---|---|---|---|
| GPT 5 | 0.0% | 0.0% | 0.0% | 0.0% | 0.0% |
| GPT 5 Mini | 0.4% | 0.0% | 0.0% | 0.0% | 0.0% |
| Gemini 2.5 Pro | 0.0% | 0.0% | 0.0% | 0.0% | 0.0% |
| GPT-OSS 120B | 0.4% | 0.0% | 0.7% | 0.5% | 0.0% |
| Qwen3-Next Thinking | 0.0% | 0.0% | 1.3% | 0.0% | 0.0% |
| GPT-OSS 20B | 0.4% | 0.0% | 2.7% | 0.5% | 0.0% |
| Claude Sonnet 4 | 0.4% | 0.0% | 0.0% | 0.0% | 0.0% |
| Gemini 2.5 Flash | 0.0% | 0.0% | 6.7% | 0.5% | 0.0% |
| Qwen3-Next | 0.4% | 0.0% | 0.0% | 0.5% | 0.0% |

## D    DISCLOSURE OF LLM USAGE

Apart from the evaluation of results and refining the natural language statements in RegEx as described in Section 3, LLMs were used to polish the writing of this paper.

Table 11: Total number of unknown cases by domain and model for Formalization Judge when given the oracle specifications.

| Model | GSM | SQL | FOL | RegEx | Coq |
|---|---|---|---|---|---|
| GPT 5 | 0.0% | 0.0% | 0.0% | 0.0% | 0.0% |
| GPT 5 Mini | 0.0% | 0.0% | 0.0% | 0.0% | 0.0% |
| Gemini 2.5 Pro | 0.0% | 0.0% | 0.0% | 0.0% | 0.0% |
| GPT-OSS 120B | 0.4% | 0.0% | 0.0% | 0.0% | 0.0% |
| Qwen3-Next Thinking | 0.0% | 0.0% | 0.0% | 0.0% | 0.0% |
| GPT-OSS 20B | 0.4% | 0.0% | 0.0% | 0.0% | 0.0% |
| Claude Sonnet 4 | 0.0% | 0.0% | 0.0% | 0.0% | 0.0% |
| Gemini 2.5 Flash | 0.0% | 0.0% | 1.3% | 0.0% | 3.2% |
| Qwen3-Next | 0.0% | 0.0% | 0.0% | 0.0% | 0.0% |

