# OpenReview forum: "RESpecBench: How reliable is LLM-as-a-judge? Rigorous Evaluation of Specification Generation with Automated Verification"
_ICLR.cc/2026/Conference — Submitted to ICLR 2026_

### Official Review · Reviewer_kBTy · 2025-10-31

**Soundness:** 2
**Presentation:** 3
**Contribution:** 2
**Rating:** 2
**Confidence:** 3

**Summary:**

This work introduces RESpecBench, a multi-domain benchmark designed to rigorously evaluate LLMs on translating natural language into formal specifications. Its primary contribution is a suite of sound, automated verifiers that replace the unreliable "LLM-as-a-Judge" method, which the authors empirically demonstrate substantially overestimates model performance by accepting plausible but semantically incorrect specifications. The benchmark spans five domains—GSM-Symbolic+, SQL, FOL, RegEx, and Rocq Prover—and provides a crucial foundation for trustworthy autoformalization evaluation, revealing that the core challenge for LLMs lies in accurately interpreting natural language intent rather than formal reasoning.

**Strengths:**

- Unified Evaluation Framework for Diverse Formal Languages: The primary strength of this work is the successful integration of distinct, domain-specific verifiers into a single, cohesive benchmarking pipeline. Formal specification languages like SQL, FOL, and RegEx have fundamentally different semantics and require specialized tools for verification. The authors have created a unified framework that seamlessly incorporates verifiers such as Z3, DFA equivalence checkers, and CoqHammer, applying them consistently across their respective domains.
- Quantitative Demystification of "LLM-as-a-Judge" for Formal Specifications: The paper moves beyond anecdotal skepticism and provides compelling, quantitative evidence that LLM-as-a-Judge is an unreliable evaluation method for semantic correctness. By establishing a sound ground truth through automated verifiers, the authors can precisely measure the discrepancy.

**Weaknesses:**

- Limited Novelty of Core Finding: The primary result—that LLM-as-a-Judge is an unreliable evaluator—is increasingly well-known and has been demonstrated in other contexts (e.g., code generation, preference judging). While applying this critique to the specific domain of specification generation is valid, it may not be considered a highly novel finding for the community.
- Limited Technical Contribution in Verification: The benchmark's main technical strength, the verifiers, are not contributions of this work. The authors explicitly state they are integrating existing tools. The core technical insight appears to be the selection of domains that are "either decidable or prover-friendly in practice," which, while pragmatic, is a curation effort rather than a novel technical advancement.
- Scalability and Domain Selection Concerns: The choice of domains raises questions about the long-term scalability and generality of the approach. By focusing on simpler, decidable fragments (like a formally specified subset of SQL or CoqHammer-compatible Rocq problems), the benchmark may avoid the more complex, undecidable problems that pose the greatest challenge for real-world autoformalization. This limits the scope of the benchmark's applicability. As noted in the paper: "RESpecBench targets domains that are either decidable or prover-friendly in practice," which inherently restricts its challenge level

**Questions:**

Concerning the above mentioned weakness, my main question is: What systematic methodologies can guide the construction of autoformalization benchmarks to progressively incorporate undecidableor highly complex problem domains while maintaining rigorous and automated evaluation?​

---

### Official Review · Reviewer_r3No · 2025-11-01

**Soundness:** 2
**Presentation:** 2
**Contribution:** 2
**Rating:** 4
**Confidence:** 4

**Summary:**

This paper aims to evaluate the capability of LLMs in generating formal specifications from natural language descriptions. The authors complement prior work by introducing a new benchmark, RESpecBench, which spans five distinct domains, each equipped with an automated and sound verifier. Using this benchmark, the paper systematically examines both specification generation quality and the reliability of LLM-as-a-Judge evaluations. Experimental results show that across all models, true specification accuracy remains around 50–55%, while LaaJ-based evaluations consistently overestimate correctness, exhibiting high false acceptance rates.

**Strengths:**

1. The research question about evaluating LLMs on formal specification generation is both promising and highly relevant to the growing field of automated verification and formal reasoning. It addresses a core challenge in bridging natural language and symbolic formalism.
2. The proposed benchmark covers multiple semantically diverse domains (e.g., mathematics, SQL, logic, and theorem proving) and evaluates a wide range of state-of-the-art LLMs.

**Weaknesses:**

1. The technical novelty is limited. The core contributions largely involve repackaging and integrating existing datasets and verifiers into a unified benchmark format. While this consolidation is useful for the community, the methodological innovations are relatively minor.
2. The title and introduction somehow overstate the relationship between this paper and the broader concept of LLM-as-a-Judge. This paper builds a benchmark for validating natural-language-to-formal-specification translation, rather than conducting a general study of LaaJ as a methodology. Therefore, the conclusions should be explicitly scoped to this context rather than generalized to all LaaJ use cases.

**Questions:**

1. If the absolute reliability of LLM-as-a-Judge scores is low, what about their relative reliability? In other words, can LaaJ still provide consistent rankings of models (e.g., Model A judged higher than Model B), even if the absolute scores are inflated? Such a finding would still make LaaJ partially useful for comparative evaluation.
2. In Table 4, are the same models used both as specification generators and as judges? If so, could self-evaluation biases affect the results? It would strengthen the paper to clarify whether different models were used for generation and judging, or whether cross-model judging experiments were conducted.

---

### Official Review · Reviewer_XvAe · 2025-11-01

**Soundness:** 2
**Presentation:** 2
**Contribution:** 2
**Rating:** 2
**Confidence:** 3

**Summary:**

This paper focuses on verifying the semantic correctness of formal specifications generated by large language models (LLMs) from natural language (NL). It introduces **RESpecBench**, a multi-domain benchmark with 707 tasks across five domains (GSM-Symbolic+, SQL, FOL, RegEx, Rocq Prover), each paired with a reliable automated verifier. The paper evaluates 9 SOTA LLMs on RESpecBench and compares them with *LLM-as-a-Judge* (formal and equivalence judges), aiming to assess the reliability of LLM-generated formal specifications and the drawbacks of LLM-as-a-Judge.

**Strengths:**

- Constructs a multi-domain benchmark (RESpecBench) covering diverse formal reasoning tasks, providing a structured framework for evaluating NL-to-formal-specification capabilities.
- Systematically analyzes the unreliability of LLM-as-a-Judge, highlighting its limitations in overestimating or underestimating specification correctness.

**Weaknesses:**

- **Ambiguous Definition**: Fails to clearly distinguish between verifying the *semantic consistency of the informal-formal transformation process* and verifying the *equivalence between the model-generated formal specification and the oracle formal specification*. For example, in SQL tasks, the work focuses on result set equivalence between model-generated and oracle SQL but ignores the reasoning gap in the informal-formal transformation. Take the example: the informal query is Query the names of all users whose age is >30 and department is ‘R&D’. The oracle formal SQL is `SELECT name FROM users WHERE age>30 AND department='R&D'` — here, the model needs to *reason* to determine that the query should be executed on the “users” table (a piece of implicit knowledge not directly stated in the informal query). The work only checks if the model-generated SQL returns the same result set as the oracle SQL, but does not evaluate whether the model correctly navigated this informal-formal reasoning gap (e.g., what if the model incorrectly queries from a “employees” table but coincidentally returns the same result set? The current setup would still mark it as correct, despite a flawed transformation process).
- **Flawed Notion of Theorem Equivalence**: The definition of theorem equivalence is unclear. A question: are all true theorems equivalent (e.g., are  \(1+1=2\) and \(2+2=4\) equivalent) ?
- **Limited Benchmark Significance**: RESpecBench relies on oracle-based evaluation, which cannot guarantee the model’s semantic consistency in real-world scenarios without oracles. It does not provide a domain-general checker for NL-to-formal transformation without oracle references.
- **Unjustified Domain Selection**: The five domains in RESpecBench only share the trait of being “formal” but lack a coherent rationale for their combination, making the benchmark’s design appear arbitrary.

**Questions:**

1. Could you clarify the exact scope of “semantic correctness” in this work—does it target the informal-formal transformation process or the equivalence between model-generated and oracle formal specifications? Please provide explicit definitions and boundaries.
2. How do you define and verify “theorem equivalence” in domains like FOL and Rocq Prover? Please elaborate on the criteria to avoid misinterpreting logically distinct true theorems as equivalent.
3. What is the practical utility of RESpecBench in real-world scenarios where oracles are unavailable? How can this work be extended to develop a domain-general checker for NL-to-formal transformation without oracle dependencies?
4. What is the rationale behind selecting the five specific domains (GSM-Symbolic+, SQL, FOL, RegEx, Rocq Prover) beyond their shared “formality”? Please explain how their combination advances the understanding of LLM-generated formal specification reliability.

---

### Official Review · Reviewer_KFKc · 2025-11-03

**Soundness:** 2
**Presentation:** 2
**Contribution:** 2
**Rating:** 2
**Confidence:** 2

**Summary:**

This paper introduces RESpecBench, a benchmark for evaluating natural language to formal specification translation with sound automated verifiers across five domains: GSM-Symbolic+, SQL, First-Order Logic, Regular Expressions, and Rocq Prover. The key contribution is demonstrating that LLM-as-a-Judge systematically overestimates specification correctness, with Formalization Judge showing 80.9% false acceptance rate and 21.9% inflation. While the paper addresses an important problem and provides valuable empirical evidence against relying on LLM-as-a-Judge for specification evaluation, it suffers from significant limitations including severely limited dataset scale (only 707 total examples, with just 31 for Rocq Prover), extremely high Unknown verdict rates in SQL (34-50%) and Rocq (83-93%), minimal analysis of failure modes, and no investigation of whether better judge design could improve reliability. The benchmark is more of a demonstration of verification infrastructure than a comprehensive dataset.

The empirical findings about LLM-as-a-Judge unreliability are valuable, but the benchmark's utility as a lasting resource is questionable given its size and coverage limitations.

**Strengths:**

1. The use of formal verification tools including Z3, Polygon, DFA equivalence, and CoqHammer provides mathematical soundness guarantees that are clearly lacking in existing work

2. The introduction of FAR, FRR, and Inflation metrics provides interpretable measures of judge reliability. The distinction between Formalization Judge (checking NL->Spec) and Equivalence Judge (checking Spec<->Spec) is methodologically sound and reveals an important asymmetry: Formalization Judge over-accepts plausible but wrong specifications, while Equivalence Judge is overly conservative and rejects correct ones.

**Weaknesses:**

A)  The limited size of the dataset is my biggest concern with this work. The total dataset contains only 707 examples across five domains, with extremely imbalanced distribution. This scale limitation has serious implications. With only 707 examples, this benchmark cannot be used to train or fine-tune models for specification generation, severely limiting its utility as a benchmark dataset. The paper only uses it for evaluation, which is valuable but doesn't match what the research community typically expects from a benchmark resource.

B) The verification infrastructure has severe completeness problems that undermine the claimed automation and efficiency. The SQL domain shows 34.2%-50.0% Unknown rate across models according to Table 8, meaning the verifier cannot decide on up to half of SQL specifications. With only 76 SQL examples total, a 34-50% Unknown rate means you have only 38-50 decidable cases per model, which is extraordinarily small for drawing conclusions.

C) While the paper demonstrates convincingly that judges fail, it provides almost no analysis of why they fail or what kinds of mistakes they make. Section 3.2 mentions that typical misses include constraint details such as NULL field handling for SQL, but this is the only concrete failure example in the entire paper. There is no systematic error analysis categorizing judge failures

D) The paper demonstrates that LLM-as-a-Judge fails but doesn't explore whether better judge design could help. The prompts in Appendix B are simple scoring rubrics with examples, but the paper doesn't report whether alternative approaches were tested.

E) The paper claims verifiers provide ground truth but never validates this against human judgment. There is no human-verifier agreement study showing whether human experts agree with verifier verdicts, no exploration of whether there are cases where verifiers say Refuted but humans would accept the specification as reasonable, and no analysis of whether Unknown verdicts are actually unknowable or just ATP limitations.

**Questions:**

1. Why does the benchmark contain only 707 examples? Is this scale limitation due to verification infrastructure costs, manual curation effort that scales poorly, or fundamental limitations in available dataset sources?

2. The Rocq Prover domain shows 83-93% Unknown rate with only 31 examples, which raises the question of whether this domain is actually usable for evaluation. Why include it rather than expanding other domains or adding new ones that might have better verification tooling?

3. Have you checked whether the evaluation models were trained on GSM8K, FOLIO, or other source datasets used to construct RESpecBench? Data contamination could affect both generation and judge performance, and this should be explicitly addressed.

4, Did you experiment with prompt engineering to reduce FAR and FRR rates? Even simple additions to the judge prompts like "be especially careful about constraint handling and edge cases," might help

---

### Meta-Review · Area_Chair_23Sq · 2026-01-06

**Summary:**

KFKc: Too small and imbalanced benchmark (707 total, 31 Rocq), very high unknown rates (SQL 34–50%, Rocq 83–93%) weaken “automated ground truth”, little failure-mode analysis, no attempt to improve judge design, no human–verifier validation, raises contamination + “why include Rocq?” concerns.

XvAe: Scope and definitions unclear (NL to formal semantic fidelity vs oracle-equivalence), oracle/result-set equivalence can miss real NL to formal reasoning errors, “theorem equivalence” notion under-specified, limited utility without oracles, domain choice rationale seems arbitrary.

r3No: Limited novelty (mostly integration), framing overclaims general lessons about LLM-as-a-Judge beyond this setting, wants checks on relative ranking reliability and possible self-judging/cross-model judging bias.

kBTy: Main finding not very novel, verification is integration not new methods, benchmark may not scale or generalize because it targets decidable and prover-friendly fragments, asks for a principled path to include harder and undecidable domains while keeping rigorous evaluation.

**Reviewer Concerns:**

There were no rebuttals submitted

**Reviewer Scores:**

No change as there were no rebuttals

---

### Decision · Program_Chairs · 2026-01-26

Reject